# Retrospective Study of Appropriate Primary Prevention in Postmenopausal Women Presenting with a Major Adverse Cardiovascular Endpoint (MACE)

**DOI:** 10.3390/pharmacy10050105

**Published:** 2022-08-26

**Authors:** Nicole E. Cieri-Hutcherson, Aleksandra Lomakina, Maya R. Chilbert

**Affiliations:** School of Pharmacy and Pharmaceutical Sciences, University at Buffalo, Buffalo, NY 14214, USA

**Keywords:** primary prevention, MACE, major adverse cardiovascular endpoint, postmenopausal, menopausal, women

## Abstract

Background: Postmenopausal women may be at an increased risk for cardiovascular events. The postmenopausal transition represents a key time for implementation of preventative strategies to reduce the risk of cardiovascular disease. The objective of this study was to evaluate the appropriate use of primary prophylaxis of cardiovascular disease in this population and to determine if an opportunity exists for improvement in primary prevention prescribing. Methods: A single-center, retrospective study was conducted of postmenopausal women aged 45–60 years between 1 October 2019 and 30 April 2021 with a diagnosis of a new major adverse cardiovascular event (MACE). This study was approved by the University at Buffalo Institutional Review Board. Results: After application of inclusion and exclusion criteria, 231 patients were included and analyzed. Median age was 55 years; 66.6% white; median body mass index was 30.11 kg/m^2^; 30.3% history of diabetes; 51.1% current smokers; 82.3% with a primary care provider (PCP); 97.6% insured. Patients with diabetes, current smokers, and those without a PCP were more likely to have inappropriate primary prevention use than patients without diabetes, non-smokers, and with a PCP, respectively (78.7% vs. 51.3%, *p* = 0.0002; 57.6% vs. 42.4%, *p* = 0.0177; 73.7% vs. 56.0%, *p* = 0.0474). Specifically, current smokers, and those with diabetes had significantly more inappropriate use of aspirin and statins for primary prevention than non-smokers and patients without diabetes. Conclusions: This study observed the use of appropriate primary prevention therapies in postmenopausal women and found that an opportunity may exist to improve prescribing appropriate primary prevention therapies for certain groups, most notably in postmenopausal women with diabetes, smokers, uninsured, and those without a PCP.

## 1. Introduction

### 1.1. Background

Cardiovascular disease continues to be the leading cause of mortality in the United States (US) [1]. According to the Center for Disease Control (CDC), the number of deaths due to heart disease in 2020 approached 700,000. Preventing cardiovascular disease-related deaths requires the promotion of healthy lifestyle measures, as well as the addition of preventative pharmacotherapies. The American College of Cardiology and the American Heart Association (ACC/AHA) recommend several interventions for primary prevention of myocardial infarction (MI) and stroke in adults that have a high risk of cardiovascular mortality, as determined by the Atherosclerotic Cardiovascular Disease (ASCVD) score [2]. These interventions include both non-pharmacologic and pharmacologic measures. The ASCVD score determines the risk of a major adverse cardiovascular event (MACE), such as a stroke or MI, in the next 10 years. 

Non-pharmacologic therapy to reduce the risk of a MACE includes at least 150 min per week of moderate-intensity activity or 75 min per week of vigorous activity according to the ACC/AHA [2]. Individuals that are obese with a body mass index (BMI) of ≥30 kg/m^2^ should have a goal weight loss of 5–10% from those initial weight through lifestyle interventions. Smoking cessation is also highly recommended to decrease cardiovascular risk.

Although the use of pharmacotherapy for secondary prevention of cardiovascular disease has well established guidance, pharmacotherapy for primary prevention of cardiovascular disease has been controversial. Low-dose aspirin is recommended by ACC/AHA for primary prevention in adults who are 40–70 years old with an ASCVD score greater than 10% and not at an increased risk of bleeding [2]. The ACC/AHA recommends a moderate intensity statin and lipid panel monitoring for adults with an ASCVD score greater than 7.5% for primary prevention [3]. Adults 40–75 years old who are diagnosed with diabetes have an indication for a moderate to high intensity statin unless contraindicated. The ACC/AHA recommends that adults who have an ASCVD score of 10% or more with a systolic blood pressure of greater than 130 mmHg and/or a diastolic blood pressure of greater than 80 mmHg be started on first-line antihypertensive pharmacotherapy for primary prevention [4]. 

There have been few studies conducted with appropriate enrollment of women assessing primary prevention of a MACE [5]. Many early studies of primary prevention either did not include, or under-enrolled, women, limiting external validity. Additionally, the majority of existing literature focuses on the use of aspirin for primary prevention. The Women’s Health Study (WHS) was one of the first to provide data on this patient population. Findings from the WHS suggest that there may be potential sex differences in primary prevention of MI and stroke [5]. Three more recent studies have begun to fill the gap in knowledge regarding the use of primary prophylaxis for prevention of cardiovascular disease. The Aspirin to Reduce Risk of Initial Vascular Events (ARRIVE) trial enrolled 12,546 participants (one- third women) [6]. Participants received either aspirin 100 mg po daily or placebo for primary prophylaxis. There was no sex difference noted on the prevention of composite cardiovascular outcomes. However, female participants that were randomized to receive aspirin had a significant increase in risk of gastrointestinal bleeding compared to placebo (HR, 2.11; 95% CI, 1.36–3.28). Another trial, A Study of Cardiovascular Events in Diabetes (ASCEND), randomized 15,480 participants with type 2 diabetes (40% women) to receive aspirin 100 mg po daily or placebo [7]. Participants receiving aspirin had a significantly reduced risk of cardiovascular events; however, as with the ARRIVE trial, there was a significant increase in risk of bleeding in participants receiving aspirin. Similarly, the Aspirin in Reducing Events in the Elderly (ASPREE) trial randomized 19,114 healthy adults (56% women) to receive either aspirin 100 mg po daily or placebo [8]. No difference was seen in the prevention of death or major cardiovascular events; but an increase in risk of major hemorrhage was seen in participants receiving aspirin. 

The AHA recognizes the transition to post-menopause as a time of significant increased risk of cardiovascular disease [9]. This risk in cardiovascular risk is a result of a complex interplay of factors, including the cessation of ovarian function. The transition to post-menopause is considered a critical time for implementing preventative strategies to reduce cardiovascular risk. The statement from the AHA also highlights the relative lack of data for primary, and secondary, prevention of cardiovascular disease and a lack of ability to make evidence-based recommendations. 

### 1.2. Objectives

Given the paucity of data in this population, the primary objective of this study was to evaluate appropriate use of primary prevention therapies among postmenopausal women who presented to the hospital with a new MACE. A secondary objective was to identify specific subgroups in the population of postmenopausal women in which an opportunity for implementation of appropriate preventative strategies exists.

## 2. Materials and Methods

### 2.1. Study Design and Setting

A retrospective cohort study of presumed postmenopausal patients presenting with a non-fatal MI or stroke admitted to a large academic medical center between 1 October 2019 and 30 April 2021 was conducted. This study was approved by the University at Buffalo Institutional Review Board (STUDY00005487 approved 21 May 2022). 

### 2.2. Participants

Patients were included if they were presumed postmenopausal women, age 45–60 years, presenting with an International Classification of Disease (ICD)-10 code of I21, I24, or I63 designating diagnosis of a new MACE. Patients who had a previous MACE per their history and physical (H&P), surgically induced menopause, or were not yet in menopause per the H&P were excluded. 

### 2.3. Variables

The 2017 ACC Hypertension in Adults guideline, 2018 ACC Blood Cholesterol guideline, and the 2018 ACC Primary Prevention guideline were used to determine appropriateness of therapy for primary prevention [2,3,4]. Patients were analyzed for inappropriate use across all three primary prevention therapy categories. Patients were placed into the category of “inappropriate use” of primary prevention therapy if any or all therapies for primary prevention were inappropriate. Patients who used all three therapies appropriately for primary prophylaxis were deemed to be “appropriate use.” Additional definitions of inappropriate use may be seen in Table 1.

To observe the differences in use of appropriate primary prevention in this population, various groups were compared: obese and non-obese; current smokers and non-current smokers; patients with a primary care provider (PCP) and those without a PCP; patients with and without diabetes; white, black, and other races; patients with and without insurance; and ages of patients. In each of these groups, “appropriate use” and “any inappropriate use” was observed. 

Demographic information collected via chart review included: age; race; body mass index (BMI); current diabetes, smoking, and PCP statuses. A 10-year ASCVD risk score was calculated at the time of admission to determine the appropriateness of statin and aspirin use for primary prevention. Patients who did not have a lipid panel were excluded from those results as the ASCVD risk score could not be calculated; the exception to this was for patients who had diabetes and were not on a statin prior to admission since these patients would be indicated for a moderate intensity statin regardless of ASCVD risk score. Patients without a diagnosis of diabetes per H&P but who have a hemoglobin A1C (HbgA1c) of ≥6.5% upon admission were defined as diabetic. All patients were presumed to be postmenopausal due to their age at presentation and based on review of H&P unless otherwise stated. 

### 2.4. Bias

All data collection was conducted by a single investigator to reduce misclassification of patient data.

### 2.5. Study Size

The time period selected was based on a change in guidelines in September 2018. A time period of one year was allowed for implementation of guidelines. The study size was based on the number of eligible patients admitted from the time of the guideline change to the time of study initiation.

### 2.6. Statistical Methods

Bivariate comparisons between appropriate use of aspirin, statins, and antihypertensive therapies and various patient characteristics, including age, race, diabetes, smoking, obesity, insurance, and current PCP status. Statistical analysis was done using the Fisher’s exact test with all *p*-values two-sided and those less than 0.05 were reported as significant. Multivariable logistic regression was completed to test the association between use of appropriate primary prevention and various patient groups. Backwards stepwise elimination was used to select variables significantly associated with inappropriate primary prevention use. Similarly, *p*-values less than 0.05 were reported as significant. Data analyses were completed using SAS, version 9.4 (SAS Institute Inc., Cary, NC, USA).

## 3. Results

### 3.1. Participants

A total of 552 patients admitted between 1 October 2019 and 30 April 2021 with a new MACE were screened for inclusion; after application of inclusion and exclusion criteria, 231 were analyzed (Figure 1). 

### 3.2. Descriptive Data

Patient demographics may be seen in Table 2. 

### 3.3. Outcome Data

A comparison of the primary objective of assessment of appropriate use of primary prevention therapies may be seen in Table 3. Categorization for inappropriate use of primary prevention therapies can be seen in Table 4.

### 3.4. Other Analyses 

Patients with diabetes, current smokers, and those without a PCP were more likely to have inappropriate primary prevention therapies compared with patients without diabetes, non-current smokers, and patients with a PCP, respectively (78.7% vs. 51.3%, *p* = 0.0002; 57.6% vs. 42.4%, *p* = 0.0177; 73.7% vs. 56.0%, *p* = 0.0474) (Table 5). Current smokers, and those with diabetes had more inappropriate aspirin use than non-current smokers, and patients without diabetes, respectively (38.0% vs. 25.2%, *p* = 0.0473; 50.8% vs. 24.0%, *p* = 0.0001) and statins (52.7% vs. 29.5%, *p* = 0.0005; 55.2% vs. 35.3%, *p* = 0.0060) (Table 4). Uninsured patients also used aspirin more inappropriately than patients with insurance (80% vs. 69.4%, *p* = 0.0361). Patients without a PCP had more inappropriate use of statin therapy compared with those that did have a PCP (56.4% vs. 38.2%, *p* = 0.0366). 

Results were confirmed via logistic regression, in which current smokers and patients with diabetes were shown to have an increased risk of inappropriate primary prevention use (OR 2.2, 95% CI: 1.2–3.9; OR 3.8, 95% CI: 1.9–7.3, respectively (Table 6).

## 4. Discussion

### 4.1. Key Results and Interpretation

An opportunity appears to exist for improvements in appropriate primary prevention prescribing, particularly for certain groups of presumed postmenopausal women. Postmenopausal women with diabetes and those who smoked appeared to have more inappropriate use of primary prevention therapies compared with other groups which may identify a target for optimal intervention. The AHA recognizes the transition to post-menopause as a time of significant increased risk of cardiovascular disease [9]. This is a critical time for implementing preventative strategies to reduce cardiovascular risk. This analysis highlights subgroups of a patient population for targeted intervention with particularly high risk for development of subsequent cardiovascular disease. 

We identified postmenopausal women who were current smokers to have more inappropriate use of aspirin and statins compared to non-current smokers; and smoking is one of the leading causes of a MACE [10]. A study conducted by Attard et al. found that current smokers had a 2.7-fold (95% CI: 1.7–4.2) increased risk of MI compared to non-smokers which highlights the need to intervene in this population if possible. A cohort study done by Jorm et al. evaluated smoking and the use of primary care services [11]. This large study found that smokers had significantly lower rates of using certain preventive services such as pap smears, immunizations, and health checks. They concluded that current smokers were less likely than others to use primary care services and may be missing out on preventative services from which they would benefit. Though they did not specifically assess primary prevention medications, this data along with the results of the present study demonstrate the great need for intervention. The pharmacist’s ability to aid with smoking cessation has been demonstrated in multiple studies to improve patients’ ability to quit smoking as well as improve patient satisfaction [12]. This would also provide an opportunity for the pharmacist to assess the patient, provide education, and intervene on other risk factor reduction such as suggesting initiation of lipid-lowering medications, anti-hypertensives, and anti-diabetic medications as needed. 

It has been reported in an evaluation of postmenopausal women that the pattern of statin prescribing was significantly higher for secondary prevention compared to primary prevention [13]. This study is in alignment with the present results and concluded that although primary care physicians are aware of cardiovascular risks, results suggested that they may not consistently implement guidelines. This can extend beyond the primary care physicians’ role and create an opportunity for pharmacist involvement. It has been shown in a meta-analysis of studies assessing pharmacist intervention for clinical management of cardiovascular risk factors that pharmacists have been able to reduce these risk factors either independently or in collaboration with a physician or nurse [14]. In fact, their impact on lipid panels was statistically significant showing a reduction in total cholesterol of −17.4 mg/L (95% CI, −25.5 to −9.2) and low-density lipoprotein cholesterol (LDL) −13.4 mg/L (95% CI, −23.0 to −3.8). 

Postmenopausal women with insurance appeared more likely to have appropriate aspirin use compared with uninsured. Uninsured postmenopausal women also appeared to be less likely to have a PCP compared with insured patients. This may be a contributing factor as to why uninsured patients do not use primary prevention therapies appropriately as they do not have a provider evaluating their need for these therapies as often as someone with a PCP may be. A cross-sectional study conducted by Liu et al. evaluated the use of aspirin for primary prevention in the uninsured patient population [15]. Out of the 1443 patients that met the criteria for aspirin therapy according to the AHA/ACC guidelines, only 17% of patients aged 50–59 years and 15% of patients aged 60–69 years were taking aspirin therapy. These results were not entirely surprising given that uninsured individuals may have less access to healthcare and would therefore be less likely to be on appropriate primary prevention therapies. However, even if patients do not have insurance and/or a PCP, many still have relationships with their community pharmacist or easy access to a community pharmacy. It has been demonstrated that student-pharmacists in the community setting are able to appropriately, and effectively, identify patients with diabetes who qualify for aspirin as well as implement a process for initiation [16]. A community pharmacy may be an ideal location to engage patients without other healthcare access in appropriate primary prevention discussions, and possibly even refer to PCP or enroll in insurance coverage.

Postmenopausal status is an additional consideration for cardiovascular health and the present study identified those who currently smoke, those with diabetes, ad those without insurance to be the least likely to be on appropriate primary prevention therapies. At baseline, patients who smoke or with diabetes are at an increased risk for a MACE, particularly without appropriate primary prevention. The ability to identify which subgroups in this population appear to use these therapies inappropriately can help practitioners better identify patient groups that may be more at risk for misuse. The community pharmacist can play a large role in this setting by identifying a need, implementing primary preventative strategies, and educating the patients [17]. Furthermore, improved communication is needed between the inpatient and outpatient setting among providers taking care of the patient to ensure that primary prevention therapies, when indicated, do not go unprescribed [17]. Patients presenting to the inpatient setting for reasons other than a MACE should be assessed for initiation of primary prevention therapies if indicated. Additionally, a PCP should be established for patients admitted to the hospital without a PCP. By preventing misuse, patients will have lower rates of adverse effects and a possible decreased risk of a MACE. 

### 4.2. Limitations

Limitations of this study include its retrospective, single-center design. Additionally, recall bias or misclassification bias may have occurred during data collection. There is also the risk of improper documentation by the provider who wrote the patient’s H&P or collected a prior to admission medication list upon admission which may have had an impact on patient selection and results. It is also important to note that patients who presented with a MACE and an HbgA1c of ≥6.5% upon admission were considered to have diabetes even if they did not have a previously documented diagnosis prior to admission. This may have altered the results in the population of patients with diabetes since these patients would not be taking the appropriate primary prevention therapy if they were not aware of their diagnosis prior to admission. An evaluation of patient adherence to medications was not conducted; as such, patients may have not been adherent to their primary prevention therapies and included in the appropriate use category incorrectly. Postmenopausal status was assumed due to patient age and H&P review. Due to the relatively small sample size, external validity is limited. 

### 4.3. Generalizability

This study highlights differences in appropriate use of primary prevention and which groups of postmenopausal, female patients may be at an increased risk of inappropriate therapy. The implications that this study has for real world practice is to make changes in the outpatient care setting, specifically in primary care. Extra care should be taken to evaluate the ASCVD risk in women who are postmenopausal since a leading cause of morbidity and mortality in this population is cardiovascular disease.

## 5. Conclusions

This single-center, retrospective study observed the use of appropriate primary prevention therapies in postmenopausal women and found that an opportunity may exist to improve prescribing appropriate primary prevention therapies for certain groups, most notably in postmenopausal women with diabetes, smokers, uninsured, and those without a PCP. Pharmacists have a unique ability to identify these patients and intervene as they are easily accessible to the community, even for patients without insurance or a PCP. 

## Figures and Tables

**Figure 1 pharmacy-10-00105-f001:**
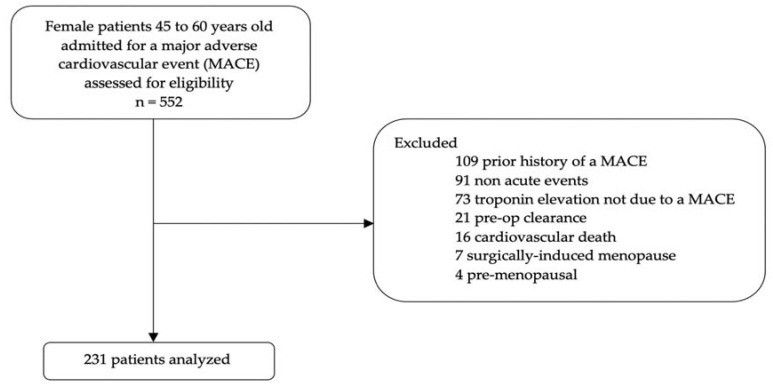
Flowchart of exclusion criteria application.

**Table 1 pharmacy-10-00105-t001:** Appropriateness definitions.

	Inappropriate Use of Primary Prevention Therapy
Aspirin	Aspirin use without indication
	Aspirin with indication but without use
	Use of aspirin 325 mg
Statin	Patients with diabetes without statin
	Patients with diabetes but on low-intensity statin
	Statin use without indication
	Statin with indication but without use
Antihypertensive	Hypertension treatment without indication
	Hypertension treatment indicated without use
	Non-first line treatment for hypertension

**Table 2 pharmacy-10-00105-t002:** Patient demographics.

Variable	Descriptionn n = 231 *
Age, median (years)	55
Race, n (%)	
White	154/231 (66.7)
Black	52/231 (22.5)
Other	25/231 (10.8)
Obese (BMI > 30), n (%)	129/229 (46.5)
Current smokers, n (%)	120/231 (51.9)
Current PCP use, n (%)	190/231 (82.3)
Patients with diabetes, n (%)	70/231 (30.3)
Patients with insurance, n (%)	206/211 (97.6)
Appropriate use (all), n (%)	86/211 (40.8)
Appropriate use of aspirin, n (%)	144/211 (68.2)
Appropriate use of statin, n (%)	127/217 (58.5)
Appropriate use of hypertension treatment, n (%)	191/231 (82.7)

BMI = body mass index, PCP = primary care physician. * n dependent on if information available for the specified variable.

**Table 3 pharmacy-10-00105-t003:** Comparison of appropriate primary prevention.

Variable	Appropriate Use n = 86	Inappropriate Use ^I^ n = 125	*p*-Value
Age, median (years) n = 231	55	55	0.5722 ^#^
Race, n (%)			0.8922 ^
White	60/86 (69.8)	83/125 (66.4)	
Black	15/86 (17.4)	28/125 (22.4)	
Other	11/86 (12.8)	14/125 (11.2)	
Obese patients, n (%)	40/86 (46.5)	69/125 (55.2)	0.2622 *
Current smokers, n (%)	35/86 (40.7)	72/125 (57.6)	0.0177 *
Current PCP use, n (%)	76/86 (88.4)	97/125 (77.6)	0.0474 *
Patients with diabetes, n (%)	13/86 (15.1)	48/125 (38.4)	0.0002 *
Patients with insurance, n (%)	86/86 (100)	120/125 (96.0)	0.0811 *

PCP = primary care physician. ^#^ Mann–Whitney U test; * Fisher’s exact test; ^ Kruskal–Wallis test. ^I^ Inappropriate use defined as any inappropriate use within the three categories of primary prevention.

**Table 4 pharmacy-10-00105-t004:** Categorization of inappropriate use of primary prevention.

Inappropriate Use of Primary Prevention Therapy	Number of Subjects (n, %)
Aspirin use without indication	26 (12.3)
Aspirin with indication but without use	41 (19.4)
Use of aspirin 325 mg	4 (1.9)
Patients with diabetes without statin	38 (17.5)
Patients with diabetes but on low-intensity statin	4 (1.8)
Statin use without indication	13 (6.0)
Statin with indication but without use	35 (16.1)
Hypertension treatment without indication	4 (1.7)
Hypertension treatment indicated without use	34 (14.7)
Non-first line treatment for hypertension	2 (0.9)

**Table 5 pharmacy-10-00105-t005:** Comparison of appropriate primary prevention use among various subgroups.

Variable	Appropriate Use of Aspirin n = 144	*p*-Value	Appropriate Use of Statins n = 127	*p*-Value	Appropriate HTN Treatment n = 191	*p*-Value
Age, median (years)	56	0.8630 ^#^	55	0.4944 ^#^	56	0.5675 ^#^
Race, n (%)		0.1359 ^		0.1109 ^		0.8256 ^
White	103 (71.5)		92 (72.4)		129 (67.5)	
Black	24 (16.7)		21 (16.5)		42 (22.0)	
Other	17 (11.8)		14 (11.0)		20 (10.5)	
Obese patients, n (%)	71 (49.3)	0.3394 *	64 (50.4)	0.6084 *	98 (51.3)	0.7172 *
Current smokers, n (%)	67 (46.5)	0.0473 *	53 (41.7)	0.0005 *	98 (51.3)	0.6709 *
Patients with diabetes, n (%)	30 (20.8)	0.0001 *	30 (23.6)	0.0060 *	53 (27.7)	0.0649 *
Patients with a PCP, n (%)	117 (81.3)	0.6815 *	110 (86.6)	0.0366 *	160 (83.8)	0.1868 *
Patients with insurance, n (%)	143 (99.3)	0.0361 *	125 (98.4)	0.6512 *	187 (97.9)	1.0000 *

HTN = hypertension, PCP = primary care physician. ^#^ Mann–Whitney U test; * Fisher’s exact test; ^ Kruskal–Wallis test.

**Table 6 pharmacy-10-00105-t006:** Predictors of inappropriate primary prevention use.

Effect	Point Estimates	95% Confidence Interval	*p*-Value
Current smokers	2.157	1.21–3.86	0.0096 ^$^
Patients with a PCP	0.482	0.22–1.03	0.0608 ^$^
Patients with diabetes	3.825	1.92–7.61	0.0001 ^$^

PCP = primary care physician. ^$^ Backwards stepwise multiple variable logistic regression analysis.

## Data Availability

Not applicable.

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
