# Peer review of "Retrospective Study of Appropriate Primary Prevention in Postmenopausal Women Presenting with a Major Adverse Cardiovascular Endpoint (MACE)"

_pharmacy, 2022, doi:10.3390/pharmacy10050105_

Round 1

Reviewer 1 Report

This manuscript has several major concerns 

Type of article line 1

Add study design to the title

Correspondence line 12

What are these numbers in brackets in abstract 

Define all abbreviations on first mentioning 

English language needs to be improved throughout the manuscript 

References are missing throughout the manuscript 

I do not find any novelty in this study and manuscript needs to be severely improved. Please add paragraph on pharmacists role in these patients and try to include studies published in last 3 years. 

Author Response

Thank you for your kind review of our article. Please find below point-by-point response to the review.

  • Type of article line 1
    • Thank you, we have edited the title to: Retrospective study of appropriate primary prevention in postmenopausal women presenting with a major adverse cardiovascular endpoint (MACE)
  • Add study design to the title
    • As above.
  • Correspondence line 12
    • Thank you for noticing this oversight. The email of the corresponding author was added.
  • What are these numbers in brackets in abstract
    • Numbers removed.
  • Define all abbreviations on first mentioning
    • Thank you. Reviewed for abbreviations on first mention and edited as appropriate.
  • English language needs to be improved throughout the manuscript
    • Significant editing the manuscript text was undertaken.
  • References are missing throughout the manuscript
    • Re-referencing was undertaken in the introduction and the discussion sections of the paper as appropriate.
  • I do not find any novelty in this study and manuscript needs to be severely improved. Please add paragraph on pharmacists role in these patients and try to include studies published in last 3 years.
    • Thank you for this observation. As such, the introduction, discussion, and conclusion were significantly revised to highlight the importance of bolstering the body of literature in this patient population and the role of the pharmacist. Literature exists on potential pharmacist roles to improve primary preventive patient care which supports our results and allows a discussion of high-risk patient identification.

Reviewer 2 Report

Dear authors,

This study is a retrospective cohort study that aims at determining if opportunity exists for improvement in primary prevention prescribing habits in postmenopausal women.

Please consider some comments.

Tittle

1.     Although the title is already long, it would be nice to add the design of the study

Material and methods

1.     This study is an observational study and should be reported according to STROBE checklist. The mat met section should be reorganize and the checklist should be added as supplementary file.

2.     L81: add the reference of the approval and the date

3.     L83: Explain diagnosis codes or add reference

4.     L101 The meaning of PCP should be given the first time it appears in the text and not line 106

5.     L88-99 It is very difficult to understand your classification. Maybe be, you could present it in a Table.

Results

1.     Figure 1: Revise the tittle of this figure. It is a flowchart and not “Exclusion criteria”.

2.     Figure 1: Use a classical flowchart. The exclusion participants should not be aligned.

Author Response

Thank you for your kind review of our article. Please find below point-by-point response to the review.

  • Although the title is already long, it would be nice to add the design of the study
    • Thank you for this observation. We agree and have edited the title to: Retrospective study of appropriate primary prevention in postmenopausal women presenting with a major adverse cardiovascular endpoint (MACE)
  • This study is an observational study and should be reported according to STROBE checklist. The mat met section should be reorganize and the checklist should be added as supplementary file.
    • Thank you for this observation. While the authors agree that the study benefited from the application of most areas of the STROBE checklist, due to the absence of a comparator cohort, only certain areas of the STROBE checklist were applied and as such the checklist not included as a supplementary file. The checklist used is linked here for your review: https://www.equator-network.org/reporting-guidelines/strobe/
  • L81: add the reference of the approval and the date
    • Thank you. We have added a study approval number and date of approval.
  • L83: Explain diagnosis codes or add reference
    • Added that these are ICD (international classification of diseases) codes and revised sentences referring to diagnosis codes.
  • L101 The meaning of PCP should be given the first time it appears in the text and not line 106
    • Thank you, PCP definition added after first appearance.
  • L88-99 It is very difficult to understand your classification. Maybe be, you could present it in a Table.
    • Table added to simplify.
  • Figure 1: Revise the title of this figure. It is a flowchart and not “Exclusion criteria”.
    • Thank you. Title edited to reflect flowchart of exclusion criteria application.
  • Figure 1: Use a classical flowchart. The exclusion participants should not be aligned.
    • The flowchart has been significantly edited and replaced. Thank you for your suggestion.

Round 2

Reviewer 1 Report

The authors have done a good job

Author Response

Thank you for your kind review of our article.

Reviewer 2 Report

Dear authors,

Your study is a retrospective cohort study and thus, in this type of study, no control group is present. thus, the study should be reported according to STROBE checklist that should be added in supplementary file.

Author Response

Thank you for your kind review of our article. The STROBE checklist for cohort studies has been completed and included as a supplementary file.